# Prognostic Value of C-Reactive Protein and Albumin in Neurocritically Ill Patients with Acute Stroke

**DOI:** 10.3390/jcm11175067

**Published:** 2022-08-29

**Authors:** Ji Hoon Jang, SungMin Hong, Jeong-Am Ryu

**Affiliations:** 1Division of Pulmonology and Critical Care, Department of Internal Medicine, Haeundae Paik Hospital, Inje University College of Medicine, Busan 48108, Korea; 2Division of Pulmonary and Critical Care Medicine, Department of Internal Medicine, Busan Paik Hospital, Inje University College of Medicine, Busan 47392, Korea; 3Department of Critical Care Medicine, Samsung Medical Center, Sungkyunkwan University School of Medicine, Seoul 06351, Korea; 4Department of Neurosurgery, Samsung Medical Center, Sungkyunkwan University School of Medicine, Seoul 06351, Korea

**Keywords:** C-reactive protein, albumin, prognosis, stroke, intensive care unit

## Abstract

We evaluated the prognostic value of C-reactive protein (CRP), albumin, CRP clearance (CRPc) and CRP/albumin ratio (CAR) in neurocritically ill patients with acute stroke. This is a retrospective, observational study. We included acute stroke patients who were hospitalized in the neurosurgical ICU from January 2013 to September 2019. The primary outcome was in-hospital mortality. A total of 307 patients were enrolled in the study. Among them, 267 (87.0%) survived until discharge from the hospital. CRP and CAR were significantly higher in non-survivors than in survivors (both *p* < 0.001). Serum albumin levels were significantly lower in the non-survivors than in the survivors (*p* < 0.001). In receiver operating characteristic curve analysis for prediction of in-hospital mortality, the area under the curve of CRP (C-statistic: 0.820) and CAR (C-statistic: 0.824) were greater than that of CRPc (C-statistic: 0.650) and albumin (C-statistic: 0.734) (all *p* < 0.005). However, there was no significant difference in the predictive performance between CRP and CAR (*p* = 0.287). In this study, CRP and CAR were more important than CRPc and albumin in predicting mortality of neurocritically ill patients with stroke. Early CRP level and CAR determination may help to predict the in-hospital mortality of these patients.

## 1. Introduction

Serum biomarkers may help predict the prognosis of critically ill patients and make early decisions for their treatment [1]. Among the numerous biomarkers, C-reactive protein (CRP) and serum albumin are commonly used predictors of morbidity and mortality in critically ill patients [2,3]. CRP can be increased in the presence of non-specific acute-phase inflammation [4,5]. Markedly increased CRP levels may be associated with poor outcomes in critically ill patients [6] and changes in CRP concentrations have been associated with the outcomes of critically ill septic patients [7]. In addition, malnutrition is associated with poor outcomes in intensive care unit patients (ICU) [8,9,10]; albumin may reflect the nutritional state of critically ill patients and a low level of serum albumin may be associated with a malnutritional state in these patients [11,12]. Therefore, serum albumin levels may help to predict the clinical outcomes of critically ill patients [13,14]. Consequentially, the CRP/albumin ratio (CAR) can reflect inflammation as well as malnutrition [15,16]. Therefore, CAR may be a useful biochemical marker for predicting clinical outcomes in critically ill patients [17].

High CRP levels may reflect the progression of vascular disease [4,5] and may be associated with poor neurological outcomes in patients with ischemic stroke [5,18]. On the other hand, low serum albumin is associated with poor prognosis and mortality in aneurysmal subarachnoid hemorrhage [19]. Therefore, high CRP levels and low serum albumin may be also associated with poor neurological outcomes in neurocritically ill patients with stroke [19,20,21,22].

However, it is unclear whether absolute CRP levels and their changes are important, and whether CAR is also associated with the clinical prognosis of neurocritically ill patients with stroke. Therefore, in the present study, we evaluated the prognostic value of CRP, albumin, CRP clearance (CRPc), and CAR in neurocritically ill patients with acute stroke.

## 2. Methods

### 2.1. Study Population

This is a retrospective, single-center, observational study. Adult patients who were admitted to the neurosurgical ICU in our tertiary hospital (Samsung Medical Center, Seoul, Republic of Korea) from January 2013 to September 2019 were eligible. This study was approved by the Institutional Review Board (IRB) of Samsung Medical Center (IRB approval number: SMC 2020-09-082). The requirement for informed consent was waived by IRB due to its retrospective nature. We included patients (1) who were hospitalized in the neurosurgical ICU, (2) those with acute stroke, and (3) those with initial levels of CRP and serum albumin collected within 12 h after ICU admission and with follow-up levels. Of these patients, we excluded patients (1) who were aged below 18 years, (2) those who underwent neurosurgery and surgical intervention within 7 days after ICU admission, (3) those with acute infectious disease, (4) those with severely neurodegenerative diseases and chronic neurological abnormality, (5) those with insufficient medical records, (6) those with a ‘do not resuscitate’ order, and (7) those who were admitted to departments other than neurosurgery.

### 2.2. Definitions and Endpoints

In this study, baseline characteristics of comorbidities, ICU management, and laboratory data were collected retrospectively using Clinical Data Warehouse, which was designed for investigators to search and retrieve de-identified medical records from the electronic archives. Acute stroke is defined as the acute onset of focal neurological findings in a vascular territory or altered consciousness as a result of an underlying cerebrovascular disease such as cerebral infarction, intracranial hemorrhage other than trauma, or subarachnoid hemorrhage. The concentration of CRP and serum albumin levels were collected from ICU admission to 7 days. Baseline CRP was defined as the maximal level of CRP within 72 h from ICU admission. Baseline albumin was defined as a minimal level of serum albumin within 72 h after ICU admission. The CAR was defined as the percentage of the ratio of baseline CRP to albumin (100 × baseline CRP/baseline albumin). CRP kinetics was expressed as ΔCRP concentrations, which are the differences between baseline and subsequent measurement (the lowest serum CRP level from the 4th to the 7th day after admission). CRPc was calculated as the percentage of ΔCRP over the baseline CRP level [7]. Serum CRP levels were measured using immunoturbidimetric assays (CRPL3, Roche Diagnostics, Indianapolis, IN, USA) with a lower reference limit of 0.3 m/dL [23]. Acute Physiology and Chronic Health Evaluation (APACHE) II scores were calculated with the worst values recorded during the initial 24 h after the ICU admission [24,25]. This score ranges from a minimum of 0 to a maximum of 71; increasing score is associated with an increasing risk of hospital death. If the patient was intubated, the verbal score of the Glasgow Coma Scale (GCS) was estimated using the eye and motor scores as described previously [26]. The primary outcome was in-hospital mortality.

### 2.3. Statistical Analyses

Continuous variables were presented as means plus or minus standard deviations, and categorical variables were represented as numbers with subsequent percentages. Data comparison was carried out using Student’s *t*-test for continuous variables, whereas the Chi-square test or Fisher’s exact test was used for categorical variables. We assessed the predictive performance of CRP, CRP variants, and albumin using the areas under the curve (AUCs) of the receiver operating characteristic (ROC) curves for sensitivity vs. 100-specificity. We compared AUCs using the nonparametric approach published by DeLong et al. [27] for two correlated AUCs. Clinically relevant variables, including CRP, CRPc, CAR, albumin, age, sex, comorbidities, habitual risk factors, classification of stroke subtypes, APACHE II score on ICU admission, and ICU management were subjected to multiple logistic regression analyses to obtain statistically meaningful predictors. Due to the high proportion of malignant patients in this study, it was not possible to exclude all cancer patients from the study. Therefore, multiple logistic regression and subgroup analyzes according to the presence of malignancies were performed, because CRP can also be increased by cancer itself. Adequacy of the prediction model was determined using the Hosmer-Lemeshow test, along with the areas under the curve (AUC). All tests were two-sided and *p* < 0.05 was considered statistically significant. All data were analyzed using R Statistical Software (version 4.2.1; R Foundation for Statistical Computing, Vienna, Austria).

## 3. Results

### 3.1. Baseline Characteristics and Clinical Outcome

A total of 307 patients were enrolled in the study (Figure 1). Among them, 267 (87.0%) survived until discharge from the hospital. The patients’ median age was 59.5 ± 16.2 years. One hundred and forty-seven patients (47.9%) were males. Hypertension (47.9%) and malignancy (31.3%) were the most common comorbidities in the study population. Intracranial hemorrhage (55.4%) was the most common cause of ICU admission. APACHE II score on ICU admission was higher in the non-survivors than in the survivors (*p* < 0.001). GCS on ICU admission was lower in the non-survivors than in the survivors (*p* < 0.001). In addition, mechanical ventilation, more than one hyperosmolar agent, and vasopressor were more commonly used in the non-survivors than in the survivors. A comparison of baseline characteristics of both survivors and non-survivors is presented in Table 1.

### 3.2. Relationship between CRP, CRP Variants, and Albumin and Clinical Outcomes

Levels of serum CRP, albumin, and CAR along with ΔCRP, and CRP_C_ were compared between the survivors and the non-survivors (Table 2). CRP and CAR were significantly higher in non-survivors than in survivors (both *p* < 0.001). Also, in the subgroup analysis in patients with neurological deficit with a GCS of less than 13, CRP and CAR were significantly higher in non-survivors (*p* = 0.019 and *p* = 0.036, respectively). In addition, CRP and CAR were also significantly higher in non-survivors in a subgroup analysis of patients who underwent mechanical ventilation (both *p* = 0.004). Serum albumin levels were significantly lower in non-survivors than in survivors (*p* < 0.001). However, there were no significant differences in ΔCRP and CRP_C_ between the two groups (*p* = 0.261, *p* = 0.701, respectively). In multivariable analysis, CRP (adjusted odds ratio [OR]: 1.41, 95% confidence interval [CI]: 1.05–1.91), APACHE II score on ICU admission (adjusted OR: 1.09, 95% CI: 1.01–1.17), mechanical ventilation (adjusted OR: 7.74, 95% CI: 2.48–30.03), use of vasopressor (adjusted OR: 4.46, 95% CI: 1.15–18.25), use of mannitol (adjusted OR: 0.24, 95% CI: 0.07–0.67), and use of more than one hyperosmolar agent (adjusted OR: 4.21, 95% CI: 1.67–11.33) were significantly associated with in-hospital mortality (Hosmer–Lemeshow Chi-squared = 5.074, *df* = 8, *p* = 0.680) with AUCs of 0.936 (95% CI 0.902–0.970) (Table 3). In ROC curve analysis for prediction of in-hospital mortality, the AUCs of CRP (C-statistic: 0.820, 95% CI: 0.772–0.863) and CAR (C-statistic: 0.824, 95% CI: 0.776–0.866) were greater than that of CRPc (C-statistic: 0.650, 95% CI: 0.592–0.704) and albumin (C-statistic: 0.734, 95% CI: 0.679–0.784) (all *p* < 0.05). However, there was no significant difference in the predictive performance between CRP and CAR (*p* = 0.287) (Figure 2).

There were no significant differences in CRP and CAR between stroke patients with malignancy and those without malignancy (Appendix A). In addition, in-hospital mortality was similar between these two groups. Although CRP level and CAR were different between stroke subtypes (Appendix A), patterns of ROC curves for predicting in-hospital mortality were similar for the patients with intracranial hemorrhage, the patients with subarachnoid hemorrhage, and total patients (Appendix A).

## 4. Discussion

In the present study, we investigated the prognostic value of CRP, albumin, CRPc, and CAR in neurocritically ill patients with acute stroke. The major findings were as follows: First, there were significant differences in CRP, CAR, and albumin between survivors and non-survivors. However, there were no significant differences in ΔCRP and CRPc between the two groups. Second, in the multivariable analysis, CRP, APACHE II score on ICU admission, mechanical ventilation, use of vasopressor, use of mannitol, and use of more than one hyperosmolar agent were significantly associated with in-hospital mortality. Finally, CRP was demonstrated as a useful biomarker for predicting in-hospital mortality. However, there was no significant difference in the predictive performance between CRP and CAR. Therefore, CRP and CAR may be equally helpful to predict the prognosis of neurocritically ill patients with stroke and to make early decisions for their treatment.

Proinflammatory cytokines may be associated with neurological deterioration in patients with stroke [28]. CRP is an acute-phase reactant that synthesized by hepatocytes and regulated by proinflammatory cytokines, especially interleukin (IL)-6 [29]. CRP levels can be obtained quickly and easily compared to proinflammatory cytokines including IL-6, IL-1β, and nucleotide-binding domain and leucine-rich repeat protein-3 (NLRP3), which makes CRP testing superior to proinflammatory cytokines. In addition, CRP levels positively correlated with infarction volume compared with other laboratory tests and proinflammatory cytokines [30]. As a result, CRP plays an important role in the progression of cerebral tissue injury [30].

High CRP levels are associated with long-term poor functional outcomes in patients with ischemic stroke [23]. Acute local inflammation and changes in inflammatory cytokines levels develop in patients with ischemic brain injury due to arterial occlusion [18,23]. CRP levels can reflect the extent of cerebral infarction [18]. Ischemic stroke patients with early CRP elevation have an increased risk of death and cardiovascular mortality [18]. In addition, high CRP levels are associated with delayed cerebral ischemia and vasospasm after subarachnoid hemorrhage [23,31,32]. Therefore, in the early stage, high levels of CRP may be associated with clinical outcomes in patients with stroke. However, it is unclear whether absolute CRP levels and their change are important. In septic patients, rapidly decreased CRP levels have been associated with favorable clinical outcomes [7]. However, in this study, the change of CRP was not associated with clinical outcomes of neurocritically ill patients with stroke. Therefore, the absolute value of CRP may be more important in predicting mortality than its change and may be associated with vascular inflammation and disease severity in neurocritically ill patients with stroke. 

Serum albumin levels may be associated with nutritional state and clinical outcomes in critically ill patients. In addition, as an inflammatory response, albumin levels can be reduced due to a decreased hepatic synthesis and an increased vascular permeability. Similarly, in patients with intracranial hemorrhage, serum albumin may be associated with acute inflammatory response and the severity of intracranial hemorrhage. [33]. In the present study, serum albumin levels were significantly different between the survivors and non-survivors. However, the absolute difference was small between the two groups. Therefore, in this study, the CAR predictive performance may largely be due to CRP other than serum albumin. Notably, there was no significant difference in the predictive performance between CRP and CAR. Considering the clinically relevant factors including APACHE II score on ICU admission, mechanical ventilation, vasopressor, and hyperosmolar therapy, an early CRP level may help to predict in-hospital mortality of neurocritically ill patients with stroke. 

Serum CRP level can be elevated in patients with malignancies. CRP level may be associated with disease activity (i.e., quiescence or rapid growth phase) and clinical prognosis [34,35,36]. Several studies have shown that factors such as D-dimer, fibrinogen, and lactate dehydrogenase (LDH) are also highly correlated with malignant tumor activity and prognosis. [37,38,39,40,41]. In this study, there were no statistically significant differences in serum levels of D-dimer, fibrinogen, and LDH between stroke patients with malignancies and those without malignancies (Appendix A). Therefore, the stroke patients with malignancies would have had relatively low disease activity in this study. Indeed, CRP levels were similar between two groups. In addition, in this study, malignancy itself was not an important prognostic factor in multivariate analysis.

This study has several limitations. First, this was a retrospective and observational study and some data on CRP and albumin values were missing. Second, the non-randomized nature of the registry data might have resulted in selection bias. Third, elevated CRP levels can also be observed in patients with chronic inflammatory and neurodegenerative diseases [42]. The patients with severely neurodegenerative diseases and chronic neurological abnormality were excluded, but the patients with mild or undiagnosed illnesses might not be excluded. In addition, elevated CRP levels in patients with advanced or active cancers have not been evaluated in this study. Finally, a small sample size may limit statistical power in this study. Although the present study provides valuable insights, prospective large-scale studies are needed to further confirm the usefulness of CRP, CAR, and albumin in predicting clinical outcomes of stroke patients with evidence-based conclusions.

## 5. Conclusions

In this study, CRP and CAR were more important than CRPc and albumin in predicting mortality of neurocritically ill patients with stroke. However, there was no significant difference in the predictive performance between CRP and CAR. Therefore, CRP and CAR may be equally helpful in predicting the prognosis of neurocritically ill patients with stroke and in making early decisions for their treatment. Eventually, early CRP levels and CAR may help to predict the in-hospital mortality of these patients.

## Figures and Tables

**Figure 1 jcm-11-05067-f001:**
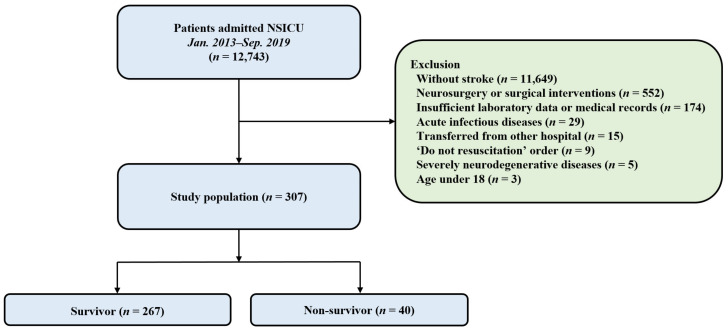
A flowchart of the study. A total of 12,743 patients admitted to the NSICU were enrolled, and 307 patients were included for the analysis. NSICU, Neurosurgical intensive care unit.

**Figure 2 jcm-11-05067-f002:**
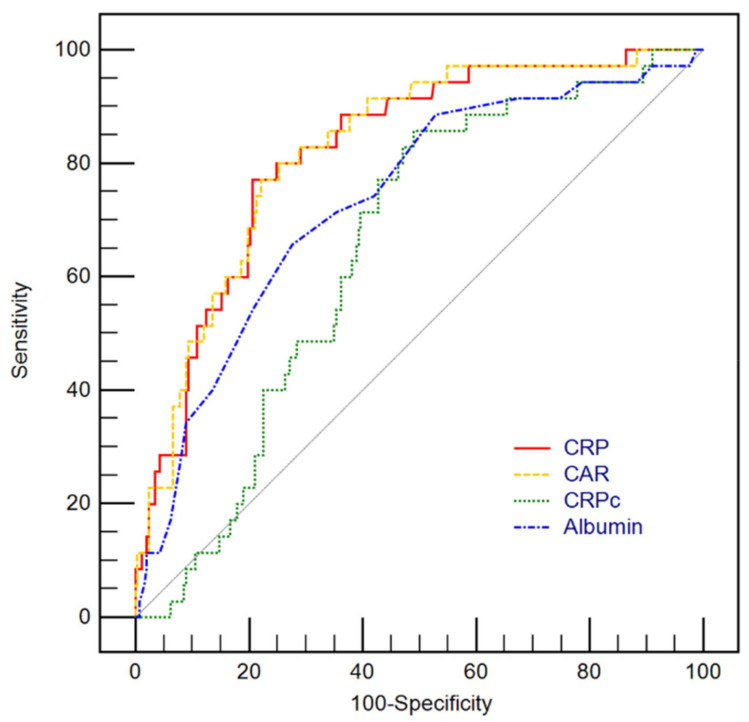
Receiver operating characteristic curves for predicting in-hospital mortality using levels of C-reactive protein (CRP), CRP to albumin ratio (CAR), CRP clearance (CRPc) along with albumin. The area under the curves (AUCs) of CRP (C-statistic: 0.820, 95% confidence interval [CI]: 0.772–0.863) and CAR (C-statistic: 0.824, 95% CI: 0.776–0.866) were greater than that of CRPc (C-statistic: 0.650, 95% CI: 0.592–0.704) and Albumin (C-statistic: 0.734, 95% CI: 0.679–0.784) with significant statistical differences (all *p* < 0.05). However, there were no statistically significant differences between AUC of CAR and CRP (*p* = 0.287).

**Table 1 jcm-11-05067-t001:** Comparison of baseline characteristics between the survivors and non-survivors.

Characteristics	Survivor (*n* = 267)	Non-Survivor (*n* = 40)	*p*
Age, years, (mean ± SD)	59.35 ± 16.52	60.77 ± 14.12	0.606
Male, *n* (%)	127 (47.6)	20 (50.0)	0.906
Comorbidities, *n* (%)			
Hypertension	123 (46.1)	22 (55.0)	0.376
Malignancy	83 (31.1)	13 (32.5)	0.999
Diabetes mellitus	42 (15.7)	8 (20.0)	0.651
Chronic kidney disease	17 (6.4)	6 (15.0)	0.107
Chronic liver disease	9 (3.4)	0 (0.0)	0.499
Habitual risk factors, *n* (%)			
Alcohol intake	82 (30.7)	15 (37.5)	0.497
Current smoker	59 (22.1)	7 (17.5)	0.650
Classification of subtype of stroke, *n* (%)			0.623
Intracerebral hemorrhage	145 (54.3)	25 (62.5)	
Subarachnoid hemorrhage	114 (42.7)	14 (35.0)	
Cerebral infarction	8 (3.0)	1 (2.5)	
APACHE II score on ICU admission (mean ± SD)	5.42 ± 5.23	11.12 ± 6.84	<0.001
GCS on ICU admission (mean ± SD)	13.75 ± 2.18	8.07 ± 3.79	<0.001
ICU management, *n* (%)			
Mechanical ventilation	93 (34.8)	35 (87.5)	<0.001
Invasive ICP monitoring	19 (7.1)	1 (2.5)	0.447
Continuous renal replacement therapy	2 (0.7)	2 (5.0)	0.143
Use of one hyperosmolar agent	120 (44.9)	12 (30.0)	0.108
Use of more than one hyperosmolar agent	81 (30.3)	23 (57.5)	0.001
Use of vasopressor	9 (3.4)	12 (30.0)	<0.001

Data are presented as either mean ± standard deviations or frequency with percentage in parentheses. SD, standard deviation; ICU, intensive care unit; APACHE, acute physiology and chronic health evaluation; GCS, Glasgow coma scale; ICP, intracranial pressure.

**Table 2 jcm-11-05067-t002:** Comparison of serum CRP and albumin levels between survivors and non-survivors.

Variable	Survivor (n = 267)	Non-Survivor (n = 40)	*p*
CRP, mg/L	3.43 ± 5.51	11.33 ± 10.08	<0.001
Albumin, g/L	3.58 ± 0.50	3.17 ± 0.50	<0.001
CAR	110.51 ± 202.81	394.03 ± 393.95	<0.001
ΔCRP	2.30 ± 4.95	1.28 ± 7.56	0.261
CRPc	−12.68 ± 271.27	3.92 ± 72.77	0.701

Data are presented as means ± standard deviations. CRP, C-reactive protein; CAR, ratio of CRP to Albumin; ΔCRP, delta CRP; CRPc, clearance of CRPs.

**Table 3 jcm-11-05067-t003:** Predicting factors for in-hospital mortality in patients assessed using logistic regression model.

Variable	Univariate Analysis		Multivariate Analysis	
	OR (95% CI)	*p*	OR (95% CI)	*p*
CRP	1.12 (1.08–1.18)	<0.001	1.41 (1.05–1.91)	0.023
CRPc	1.00 (0.98–1.00)	0.702		
Albumin	0.21 (0.10–0.42)	<0.001		
CAR	1.01 (1.01–1.03)	<0.001		
APACHEII score on ICU admission	1.15 (1.09–1.22)	<0.001	1.09 (1.01–1.17)	0.020
Use of mechanical ventilator	13.10 (5.40–39.16)	<0.001	7.74 (2.48–30.03)	0.001
Use of vasopressor	12.29 (4.80–32.64)	<0.001	4.46 (1.15–18.25)	0.032
Use of mannitol	0.53 (0.25–1.05)	0.079	0.24 (0.07–0.67)	0.011
Use of more than one hyperosmolar agent	3.11 (1.58–6.21)	0.001	4.21 (1.67–11.33)	0.003

OR, odds ratio; CI, confidence interval; CRP, C-reactive protein; CRPc, clearance of CRPs; CAR, ratio of CRP to Albumin; APACHE, acute physiology and chronic health evaluation; ICU, intensive care unit.

## Data Availability

Our data are available on Harvard Dataverse Network (http://dx.doi.org/10.7910/DVN/JG0ILD, accessed on 8 July 2022) as recommended repositories.

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
