# Peer review of "Prognostic Value of C-Reactive Protein and Albumin in Neurocritically Ill Patients with Acute Stroke"

_jcm, 2022, doi:10.3390/jcm11175067_

Round 1

Reviewer 1 Report

The authors analysed crp and albumin in their retrospective neuro ICU cohort with regard to in hospital mortality. They confirm previous results of early crp elevation in non-survivors. The information is not new. Furthermore the presentation of data does not allow separation of the different factors of crp elevation, i.e. neurological status versus artificial ventilation.

Author Response

August 21 2022

Dr. Minna Wang, Assistant Editor

Journal of Clinical Medicine

Manuscript ID: jcm-1858848

Title: Prognostic Value of C-Reactive Protein and Albumin in Neurocritically Ill Patients with Acute Stroke

Dear Dr. Minna Wang

Thank you very much for your letter and for the helpful comment from the reviewer. We appreciate the opportunity to resubmit our revised manuscript entitled “Prognostic Value of C-Reactive Protein and Albumin in Neurocritically Ill Patients with Acute Stroke”. As always, you and your editorial staff have again provided us with a comprehensive and prompt review. Many of the valuable and constructive points that the reviewers pointed out were well taken by all the authors. After going over the reviewer’s comments, my colleagues and I have performed additional investigation and made some revisions in hopes of improving our paper. The revised and added portions of the manuscript are stated in the “Response to Reviewers” and are underlined and highlighted in the revised manuscript for your convenience.

All authors contributed to the conception and interpretation of data, drafting of the manuscript, revising it critically for important intellectual content, and final approval of the manuscript. The whole manuscript or part of it, neither has been published and is not being considered for publication elsewhere in any language except as an abstract. None of the authors have any financial relationships with any company or any other bias or conflict of interest.

We believe that these findings have scientific and clinical impact and will be interesting and informative to your readers. We hope that, upon review, our study will be found to be meritorious of publication in the Journal of Clinical Medicine.

Yours sincerely,

Jeong-Am Ryu, M.D., Ph.D.

Department of Critical Care Medicine and Department of Neurosurgery, Samsung Medical Center, Sungkyunkwan University School of Medicine, 81 Irwon-ro, Gangnam-gu, Seoul 06351, Republic of Korea

Tel: 82-2-3410-6399, Fax: 82-2-2148-7088

Response to Reviewers

Reviewer #1:

The authors analysed crp and albumin in their retrospective neuro ICU cohort with regard to in hospital mortality. They confirm previous results of early crp elevation in non-survivors. The information is not new. Furthermore the presentation of data does not allow separation of the different factors of crp elevation, i.e. neurological status versus artificial ventilation.

  1. We appreciate forethoughtful comments for our study. As mentioned in the comment, there were previous studies that analyzed the relationship between stroke and CRP [Supplementary Ref: 1-14]. However, there was no study on the effect of the ratio of CRP to albumin in the group of patients in the neurosurgery intensive care unit including stroke. In addition, there has been no study on the effect of the degree of change in CRP on the prognosis of neurocritically ill patients with stroke. Indeed, in our previous study, CRP changes were associated with clinical outcome in severe septic patients. Especially, septic patients with a greater CRP clearance had a better prognosis [Supplementary Ref: 15]. Therefore, our study aimed to investigate the relationship between stroke, absolute CRP level, CRP clearance and the CRP/albumin ratio which is recently known as a good indicator in intensive care [Supplementary Ref: 16]. As a result, in present study, absolute CRP level and the CRP/albumin ratio were associated with clinical outcomes of stroke patients in ICU, but CRP clearance was not. Namely, CRP changes were not important to predict clinical outcome of stroke patients in ICU. From this point of view, the information conveyed by this study contains novel points because it includes analysis of the effects of CRP clearance and the ratio of CRP to albumin ratio on mortality as well as the effect of CRP, which was consistent with previous studies.

As your thoughtful recommendations, we statistically analyzed again for the relationship between neurological status and clinical factors such as mechanical ventilation and CRP elevation. Among neurocritically ill patients with a GCS score of less than 13 at the time of admission, CRP was significantly higher in non-survivors (p = 0.019) as well as CRP/albumin ratio (p = 0.036). In patients who underwent mechanical ventilation, CRP (p = 0.004) and CRP/albumin ratio (p = 0.004) were also significantly higher in non-survivors. We added these results in the following sentences in the Result section of the revised manuscript as below.

Line 137 – 140 in page 7: Also, in the subgroup analysis in patients with neurological deficit with a GCS of less than 13, CRP and CAR were significantly higher in non-survivors (p = 0.019 and p = 0.036, respectively). In addition, CRP and CAR were also significantly higher in non-survivors in subgroup analysis of patients who underwent mechanical ventilation (both p = 0.004).

We thank the reviewer for valuable recommendation. Addressing them fully has significantly strengthened the manuscript.

Supplementary references for response to the reviewer

  1. Chan, C.P.; Jiang, H.L.; Leung, L.Y.; Wan, W.M.; Cheng, N.M.; Ip, W.S.; Cheung, K.Y.; Chan, R.W.; Wong, L.K.; Graham, C.A.; et al. Multiple atherosclerosis-related biomarkers associated with short- and long-term mortality after stroke. Clin Biochem 2012, 45, 1308-1315. https://doi.org/10.1016/j.clinbiochem.2012.06.014.
  2. Elkind, M.S.; Tai, W.; Coates, K.; Paik, M.C.; Sacco, R.L. High-sensitivity C-reactive protein, lipoprotein-associated phospholipase A2, and outcome after ischemic stroke. Arch Intern Med 2006, 166, 2073-2080. https://doi.org/10.1001/archinte.166.19.2073.
  3. Gong, X.; Zou, X.; Liu, L.; Pu, Y.; Wang, Y.; Pan, Y.; Soo, Y.O.; Leung, T.W.; Zhao, X.; Wang, Y.; et al. Prognostic value of inflammatory mediators in 1-year outcome of acute ischemic stroke with middle cerebral artery stenosis. Mediators Inflamm 2013, 2013, 850714. https://doi.org/10.1155/2013/850714.
  4. Kanai, A.; Kawamura, T.; Umemura, T.; Nagashima, M.; Nakamura, N.; Nakayama, M.; Sano, T.; Nakashima, E.; Hamada, Y.; Nakamura, J.; et al. Association between future events of brain infarction and soluble levels of intercellular adhesion molecule-1 and C-reactive protein in patients with type 2 diabetes mellitus. Diabetes Res Clin Pract 2008, 82, 157-164. https://doi.org/10.1016/j.diabres.2008.07.006.
  5. Li, Y.M.; Liu, X.Y. Serum levels of procalcitonin and high sensitivity C-reactive protein are associated with long-term mortality in acute ischemic stroke. J Neurol Sci 2015, 352, 68-73. https://doi.org/10.1016/j.jns.2015.03.032.
  6. Liu, L.B.; Li, M.; Zhuo, W.Y.; Zhang, Y.S.; Xu, A.D. The role of hs-CRP, D-dimer and fibrinogen in differentiating etiological subtypes of ischemic stroke. PLoS One 2015, 10, e0118301. https://doi.org/10.1371/journal.pone.0118301.
  7. Makita, S.; Nakamura, M.; Satoh, K.; Tanaka, F.; Onoda, T.; Kawamura, K.; Ohsawa, M.; Tanno, K.; Itai, K.; Sakata, K.; et al. Serum C-reactive protein levels can be used to predict future ischemic stroke and mortality in Japanese men from the general population. Atherosclerosis 2009, 204, 234-238. https://doi.org/10.1016/j.atherosclerosis.2008.07.040.
  8. Ozkan, A.K.; Yemisci, O.U.; Saracgil Cosar, S.N.; Oztop, P.; Turhan, N. Can high-sensitivity C-reactive protein and ferritin predict functional outcome in acute ischemic stroke? A prospective study. Top Stroke Rehabil 2013, 20, 528-536. https://doi.org/10.1310/tsr2006-528.
  9. Park, S.Y.; Kim, M.H.; Kang, S.Y.; Suh, J.T.; Lee, W.I. Inflammatory marker expression and its implication in Korean ischemic stroke patients. Korean J Lab Med 2007, 27, 197-204. https://doi.org/10.3343/kjlm.2007.27.3.197.
  10. Roudbary, S.A.; Saadat, F.; Forghanparast, K.; Sohrabnejad, R. Serum C-reactive protein level as a biomarker for differentiation of ischemic from hemorrhagic stroke. Acta Med Iran 2011, 49, 149-152.
  11. Tu, W.J.; Zhao, S.J.; Liu, T.G.; Yang, D.G.; Chen, H. Combination of high-sensitivity C-reactive protein and homocysteine predicts the short-term outcomes of Chinese patients with acute ischemic stroke. Neurol Res 2013, 35, 912-921. https://doi.org/10.1179/1743132813Y.0000000228.
  12. Xie, D.; Deng, L.; Liu, X.D.; Li, J.M.; Zhang, Y.B. Role of high sensitivity C-reactive protein and other risk factors in intracranial and extracranial artery occlusion in patients with ischaemic stroke. J Int Med Res 2015, 43, 711-717. https://doi.org/10.1177/0300060515586246.
  13. Yeh, K.H.; Tsai, T.H.; Chai, H.T.; Leu, S.; Chung, S.Y.; Chua, S.; Chen, Y.L.; Lin, H.S.; Yuen, C.M.; Yip, H.K. Comparison of acute versus convalescent stage high-sensitivity C-Reactive protein level in predicting clinical outcome after acute ischemic stroke and impact of erythropoietin. J Transl Med 2012, 10, 6. https://doi.org/10.1186/1479-5876-10-6.
  14. Youn, C.S.; Choi, S.P.; Kim, S.H.; Oh, S.H.; Jeong, W.J.; Kim, H.J.; Park, K.N. Serum highly selective C-reactive protein concentration is associated with the volume of ischemic tissue in acute ischemic stroke. Am J Emerg Med 2012, 30, 124-128. https://doi.org/10.1016/j.ajem.2010.11.006.
  15. Ryu, J.A.; Yang, J.H.; Lee, D.; Park, C.M.; Suh, G.Y.; Jeon, K.; Cho, J.; Baek, S.Y.; Carriere, K.C.; Chung, C.R. Clinical Usefulness of Procalcitonin and C-Reactive Protein as Outcome Predictors in Critically Ill Patients with Severe Sepsis and Septic Shock. PLoS One 2015, 10, e0138150. https://doi.org/10.1371/journal.pone.0138150.
  16. Park, J.E.; Chung, K.S.; Song, J.H.; Kim, S.Y.; Kim, E.Y.; Jung, J.Y.; Kang, Y.A.; Park, M.S.; Kim, Y.S.; Chang, J.; et al. The C-Reactive Protein/Albumin Ratio as a Predictor of Mortality in Critically Ill Patients. J Clin Med 2018, 7. https://doi.org/10.3390/jcm7100333.

Reviewer 2 Report

INTRODUCTION

The author described the background and aim of this study well. 

METHODS

  1. Line 86: “CRP kinetics was expressed as ΔCRP concentrations, which are the differences between baseline and subsequent measurement (the lowest serum CRP level from 4th to 7th days after admission)”. Why did the author choose the lowest serum CRP level rather than the highest one?

  1. Line 90: APACHE II score was calculated. It should be explained what the interpretation of the APACHE II score (e.g. the higher the APACHE II score the higher the mortality risk)

RESULTS:

  1. Line 162: The author compared stroke with malignancy and without malignancy. However, the author didn’t explain the reason for this comparison in the background or method section. 
  2. Line 164: Table S2 showed a significant difference in CRP level and CAR between stroke types. Figure S2 showed the AUC of intracranial hemorrhage and subarachnoid hemorrhage. Why did the author not describe the AUC of cerebral infarction? What is the purpose of comparison of AUC between stroke types?
  3. Table 2: The number (n) of each survivor and non-survivor should be written at the top of the table.

DISCUSSION

  1. Line 191: “In septic patients, rapidly decreased CRP levels have been associated with clinical outcomes” Is it associated with poor or good outcomes? 

CONCLUSION

The author concludes the study appropriately based on the data.

Author Response

August 21 2022

Dr. Minna Wang, Assistant Editor

Journal of Clinical Medicine

Manuscript ID: jcm-1858848

Title: Prognostic Value of C-Reactive Protein and Albumin in Neurocritically Ill Patients with Acute Stroke

Dear Dr. Minna Wang

Thank you very much for your letter and for the helpful comment from the reviewer. We appreciate the opportunity to resubmit our revised manuscript entitled “Prognostic Value of C-Reactive Protein and Albumin in Neurocritically Ill Patients with Acute Stroke”. As always, you and your editorial staff have again provided us with a comprehensive and prompt review. Many of the valuable and constructive points that the reviewers pointed out were well taken by all the authors. After going over the reviewer’s comments, my colleagues and I have performed additional investigation and made some revisions in hopes of improving our paper. The revised and added portions of the manuscript are stated in the “Response to Reviewers” and are underlined and highlighted in the revised manuscript for your convenience.

All authors contributed to the conception and interpretation of data, drafting of the manuscript, revising it critically for important intellectual content, and final approval of the manuscript. The whole manuscript or part of it, neither has been published and is not being considered for publication elsewhere in any language except as an abstract. None of the authors have any financial relationships with any company or any other bias or conflict of interest.

We believe that these findings have scientific and clinical impact and will be interesting and informative to your readers. We hope that, upon review, our study will be found to be meritorious of publication in the Journal of Clinical Medicine.

Yours sincerely,

Jeong-Am Ryu, M.D., Ph.D.

Department of Critical Care Medicine and Department of Neurosurgery, Samsung Medical Center, Sungkyunkwan University School of Medicine, 81 Irwon-ro, Gangnam-gu, Seoul 06351, Republic of Korea

Tel: 82-2-3410-6399, Fax: 82-2-2148-7088

Response to Reviewers

Reviewer #2:

INTRODUCTION

The author described the background and aim of this study well.

  1. We appreciate reviewer’s comment.

METHODS

  1. Line 86: “CRP kinetics was expressed as ΔCRP concentrations, which are the differences between baseline and subsequent measurement (the lowest serum CRP level from 4th to 7th days after admission)”. Why did the author choose the lowest serum CRP level rather than the highest one?
  2. We appreciate reviewer’s thoughtful comment. As in our previous study that analyzed the clinical usefulness of procalcitonin and CRP as prognostic factors in patients with severe sepsis and septic shock (Ref), the lowest follow-up value was promoted to confirm how much CRP decreased. We evaluated whether “CRP clearance” as change of CRPs associate with clinical outcome of stroke patients in ICU. The definition of CRP clearance was applied in the same way as in our previous study.

Ref: Ryu, J.A.; Yang, J.H.; Lee, D.; Park, C.M.; Suh, G.Y.; Jeon, K.; Cho, J.; Baek, S.Y.; Carriere, K.C.; Chung, C.R. Clinical Usefulness of Procalcitonin and C-Reactive Protein as Outcome Predictors in Critically Ill Patients with Severe Sepsis and Septic Shock. PLoS One 2015, 10, e0138150.

  1. Line 90: APACHE II score was calculated. It should be explained what the interpretation of the APACHE II score (e.g. the higher the APACHE II score the higher the mortality risk)
  2. Thank you for pointing out. As your comment, we added following sentences in Methods section.

Line 93-94 in page 4: This score ranges from a minimum of 0 to a maximum of 71; increasing score is associated with an increasing risk of hospital death.

RESULTS:

  1. Line 162: The author compared stroke with malignancy and without malignancy. However, the author didn’t explain the reason for this comparison in the background or method section.
  2. Thank you for regardful pointing out. Since CRP can also be increased by cancer itself, we compared CRP and CAR between patients with malignancy and those without malignancy. As your kind recommendation, we embedded the following sentences in the Methods section.

Line 105-108 in page 5: Due to the high proportion of malignant patients in this study, it was not possible to exclude all cancer patients from the study. Therefore, multiple logistic regression and subgroup analyzes according to the presence of malignancies were performed, because CRP can also be increased by cancer itself.

  1. Line 164: Table S2 showed a significant difference in CRP level and CAR between stroke types. Figure S2 showed the AUC of intracranial hemorrhage and subarachnoid hemorrhage. Why did the author not describe the AUC of cerebral infarction? What is the purpose of comparison of AUC between stroke types?
  2. Thank you for your kind comments. In the case of cerebral infarction, the number of patients included was small and only one patient died. We judged that it would be difficult for AUC analysis to have statistical significance in patients with cerebral infarction. AUC according to stroke subtype was compared because levels of CRP and CAR might be different for each subtype, so it was expected that there would be a difference in mortality prediction. However, there was no significant difference according to subtype as a result of actual analysis as in Figure S1.

  1. Table 2: The number (n) of each survivor and non-survivor should be written at the top of the table.
  2. Thank you for thoughtful recommendation. We added number of survivors and non-survivors in Table 2.

DISCUSSION

  1. Line 191: “In septic patients, rapidly decreased CRP levels have been associated with clinical outcomes” Is it associated with poor or good outcomes?
  2. We apologize for the use of inappropriate terminology. As your recommendation, we revise the clear statement in Discussion section. In our previous study, rapidly decreased CRP levels have been associated with favorable clinical outcomes in septic patients.

CONCLUSION

The author concludes the study appropriately based on the data.

  1. We appreciate for your kind comment. We revised the Conclusions section as your recommendation.

Line 234-238 in page 12-13: In present study, CRP and CAR may be a useful biomarker for predicting in-hospital mortality. However, there was no significant difference in the predictive performance between CRP and CAR. Therefore, CRP and CAR may be equally helpful to predict the prognosis of neurocritically ill patients with stroke and to make early decisions for their treatment. Eventually, early CRP levels and CAR may help to predict the in-hospital mortality of these patients.

We thank the reviewer for valuable recommendation. Addressing them fully has significantly strengthened the manuscript.

Reviewer 3 Report

This article reports the findings of a retrospective study evaluating the levels of albumin, C-reactive protein and their ratio in patients in the neurosurgical ICU. The study found that high circulating levels of CRP and albumin were associated with the risk of mortality and hence, could be used as potential biomarkers with prognostic value. The manuscript is clearly written and the data support the conclusions. I don't have any suggestions. I congratulate the authors.

I don't have additional comments since I think the authors have presented their findings in sufficient detail and the results support the conclusions.
It is a retrospective study and the authors have identified it as such. The details about the patient's age, sex, and medical conditions are provided in sufficient detail. The population they have used is balanced for gender. Appropriate statistical tests are used during data analysis, and importantly they have identified and discussed the limitations of the study.

Author Response

August 21 2022

Dr. Minna Wang, Assistant Editor

Journal of Clinical Medicine

Manuscript ID: jcm-1858848

Title: Prognostic Value of C-Reactive Protein and Albumin in Neurocritically Ill Patients with Acute Stroke

Dear Dr. Minna Wang

Thank you very much for your letter and for the helpful comment from the reviewer. We appreciate the opportunity to resubmit our revised manuscript entitled “Prognostic Value of C-Reactive Protein and Albumin in Neurocritically Ill Patients with Acute Stroke”. As always, you and your editorial staff have again provided us with a comprehensive and prompt review. Many of the valuable and constructive points that the reviewers pointed out were well taken by all the authors. After going over the reviewer’s comments, my colleagues and I have performed additional investigation and made some revisions in hopes of improving our paper. The revised and added portions of the manuscript are stated in the “Response to Reviewers” and are underlined and highlighted in the revised manuscript for your convenience.

All authors contributed to the conception and interpretation of data, drafting of the manuscript, revising it critically for important intellectual content, and final approval of the manuscript. The whole manuscript or part of it, neither has been published and is not being considered for publication elsewhere in any language except as an abstract. None of the authors have any financial relationships with any company or any other bias or conflict of interest.

We believe that these findings have scientific and clinical impact and will be interesting and informative to your readers. We hope that, upon review, our study will be found to be meritorious of publication in the Journal of Clinical Medicine.

Yours sincerely,

Jeong-Am Ryu, M.D., Ph.D.

Department of Critical Care Medicine and Department of Neurosurgery, Samsung Medical Center, Sungkyunkwan University School of Medicine, 81 Irwon-ro, Gangnam-gu, Seoul 06351, Republic of Korea

Tel: 82-2-3410-6399, Fax: 82-2-2148-7088

Response to Reviewers

Reviewer #3:

This article reports the findings of a retrospective study evaluating the levels of albumin, C-reactive protein and their ratio in patients in the neurosurgical ICU. The study found that high circulating levels of CRP and albumin were associated with the risk of mortality and hence, could be used as potential biomarkers with prognostic value. The manuscript is clearly written and the data support the conclusions. I don't have any suggestions. I congratulate the authors.

I don't have additional comments since I think the authors have presented their findings in sufficient detail and the results support the conclusions.

It is a retrospective study and the authors have identified it as such. The details about the patient's age, sex, and medical conditions are provided in sufficient detail. The population they have used is balanced for gender. Appropriate statistical tests are used during data analysis, and importantly they have identified and discussed the limitations of the study.

R. We appreciate valuable comments. This study has several limitations. Although this study still provides valuable insight, prospective large-scale studies are needed to further confirm the usefulness of CRP, CAR, and albumin in predicting clinical outcomes of stroke patients with evidence-based conclusions. We appreciated your kind comment.

Round 2

Reviewer 1 Report

Some minor changes have been made. The study confirms the association of CRP and mortality in NICU patients. The causal importance of CRP elevation  for the often unfavorable outcome of patients with CRP elevation cannot be indentified in this retrospective analysis.